

# Computed tomography-based assessment of sphenoid sinus and sella turcica pneumatization analysis: a retrospective study

Mehmet Emin Dogan[1], Sedef Kotanlı[1], Yasemin Yavuz[2],
Dian Agustin Wahjuningrum[3] and Ajinkya M. Pawar[3,4]

[1] Department of Dentomaxillofacial Radiology, Faculty of Dentistry, Harran University, Haliliye, Şanlıurfa, Turkey
[2] Department of Restorative Dentistry, Harran University, Faculty of Dentistry, Haliliye, Şanlıurfa, Turkey
[3] Department of Conservative Dentistry, Faculty of Dental Medicine, Universitas Airlangga, Surabaya City, East Java, Indonesia.
[4] Conservative Dentistry and Endodontics, Nair Hospital Dental College, Mumbai, Maharashtra, India

Corresponding authors
Dian Agustin Wahjuningrum,
dian-agustin-w@fkg.unair.ac.id
Ajinkya M. Pawar,
ajinkya@drpawars.com

## ABSTRACT

**Background:** A preoperative three-dimensional examination of the sphenoid sinus anatomy, its pneumatization pattern, and its relevance to neighboring neurovascular constructions is crucial to preventing possible complications. In this study, the aim was to evaluate the relationship between sphenoid sinus pneumatization types and the sella turcica using computed tomography (CT).

**Methods:** CT data from 420 patients referred to the Department of Dentomaxillofacial Radiology were evaluated retrospectively. Sella pneumatization types were classified as conchal, presellar, incomplete sellar, and complete sellar, and they were evaluated. Obtained data were evaluated using the IBM SPSS 25.0 (Armonk, New York, USA) package program.

**Results:** CT images of 420 individuals, including 174 women and 246 men with a mean age of $43.87 \pm 17.58$ years, were included in the study. When the sella turcica morphologies were evaluated, the most widespread morphological type was irregularity in the posterior part of the dorsum sella, in 51.2% of cases. In addition, a statistically significant correlation was found between the pneumatization of the sphenoid sinus and the morphological types of sella ($p < 0.05$).

**Conclusion:** In this research endeavor, the predominant observation comprised the complete sellar sphenoid sinus pneumatization type, exhibiting irregularity in the posterior aspect of the dorsum sella, representing one of the sellar types. Notwithstanding, it is imperative to conduct additional investigations to establish the generalizability of the present study's findings.

## INTRODUCTION

The sphenoid sinus is an unique and intricate structure that is located posteriorly to the other paranasal sinuses near the base of the skull. The sphenoid sinus's pattern of pneumatization and the juxtaposition to various other neurovascular structures exhibit notable inter-individual differences, making them an intriguing topic for research in the area of anatomy. Notably, sinus expansion remains throughout life and reaches its peak only after puberty, which challenges the research of this structure (*Kapakin, 2016*). The diaphysis of the sphenoid bone contains the sphenoidal sinuses, also known as pneumatic cavities, which are lined with mucous membrane. It is well known that the morphology of the sphenoid sinus exhibits significant variation with regard to its size, shape, number of septa, and level of air fill. These variances might present difficulties in clinical and surgical applications, highlighting the need of having a complete grasp of the sphenoid sinus' anatomy (*Jaworek et al., 2010*). Previous research demonstrates the variability of the sphenoid sinus (*Kapakin, 2016*; *Jaworek-Troć et al., 2018*, *2019*, *2021*).

The endoscopic endonasal transsphenoidal approach (EETA) is a surgical modality that has gained widespread acceptance for the treatment of pituitary adenomas and other pathologies of the skull base. This approach is preferred due to its remarkable safety profile and a low incidence of complications, which can be further mitigated by utilizing precise anatomical knowledge. The procedure involves accessing the lesion *via* the nostrils and sphenoid sinus, using endoscopes and other specialized surgical instruments to perform the resection. Significantly, the EETA technique offers a direct, less invasive route to the target lesion in comparison to traditional open surgical procedures. This technique has been shown to deliver favorable outcomes, including reduced blood loss, shorter hospital stays, and faster recovery times, thereby benefiting both the patient and the healthcare system (*Terra et al., 2006*; *García-Garrigós et al., 2015*; *Locatelli et al., 2017*). The EETA has become an expedient technique, used during surgical operations for most parasellar and intrasellar tumor procedures (*García-Garrigós et al., 2015*). In EETA, the morphological characteristics of the sella turcica are noteworthy. The size, form, and pneumatization changes of the sella turcica, which is used in this surgical technique to reach the pituitary gland, might affect the outcome of the treatment. For preoperative planning and intraoperative navigation, an understanding of sella turcica shape is vital since it affects surgical access and visualization. Different conditions may be linked to various sella turcica anomalies, which can help with diagnosis and therapy. To improve surgical results and patient care, it is important to highlight the clinical significance of this association in EETA.

The transnasal approach, as compared to open craniotomy, is considered a less invasive technique resulting in lower morbidity and mortality rates (*Cavallo et al., 2005*). Endoscopic visualization of hard-to-reach areas is facilitated by the technique and the EETA has emerged as a preferred approach for most parasellar and intrasellar tumor procedures. In the pediatric population, EETA is preferred because it is less traumatic, has a shortened recovery time, and does not impact the anatomical and functional integrity of the skull, nor hinder growth in young patients (*de Divitiis et al., 2000*; *Al-Mujaini, Wali &*

*Al Khabori, 2009*; *Jaworek-Troć et al., 2022*). Sphenoid sinus morphology exhibits significant inter-individual variations, with differing size, pneumatization, and septation patterns leading to variations in sphenoid sinus segmentation. In some individuals, pneumatization may extend to the pterygoid process, the clivus, the major wing of the sphenoid, and occasionally the anterior clinoid process (*Hiremath et al., 2018*; *Jaworek-Troć et al., 2020*; *Mureșan et al., 2022*).

Studying sella turcica shape and sphenoid pneumatization types has therapeutic implications for many different medical specialties. Understanding these characteristics from a radiological perspective makes it easier to accurately interpret the imaging findings and discern abnormalities and diseases. Understanding sella turcica morphology in neurosurgery is essential for surgical planning, especially when it comes to the pituitary gland, lowering risks and enhancing outcomes (*Kjær, 2015*). Adenomas of the pituitary and other endocrinological conditions can be connected to anomalies of the sella turcica, making early detection and therapy attainable. The anthropological and forensic importance of these investigations extends beyond medicine, giving insights on human evolution and demographic distinctions. Additionally, sella turcica measurements in orthodontics affect how to manage craniofacial problems. The applicability of understanding sphenoid pneumatization variations and sella turcica morphology across numerous dental disciplines are what give this research its dental clinical value. It offers insights on craniofacial development in orthodontics, assisting with individualized treatment regimens. Surgical planning for operations in the midface is advantageous for oral and maxillofacial surgery (*Sathyanarayana, Kailasam & Chitharanjan, 2013*). The placement and design of dental prostheses are improved, which enhances prosthodontics. Additionally, it could provide information on how to treat temporomandibular joint disorders. Additionally assisted is accurate picture interpretation in dental radiology. Overall, this understanding improves dental therapy, resulting in greater healing and patient wellbeing.

*Zada et al. (2011)* have drawn attention to this issue in their research, underscoring the importance of thorough preoperative planning and imaging to ensure optimal surgical outcomes. Anatomical structures of the sphenoid sinus, as well as bony anatomical variations in this region can be easily visualized by computed tomographic (CT) examination. *Hammer & Radberg (1961)* classified sphenoid sinus variations into three anatomical structures in 1961, and this sphenoid sinus pneumatization classification (conchal, presellar and sellar) is still used. A few studies previously evaluated only sphenoid sinus pneumatization according to various classifications (*Wang et al., 2010*; *Cho et al., 2010*; *Tesfaye et al., 2021*; *Parameshwar Keerthi et al., 2022*).

Notably, despite the abundance of literature on the sphenoid sinus and the sella turcica, the relationship between the pneumatization types of the former and the latter has received scarce attention. Consequently, this study endeavors to bridge this gap in knowledge by investigating the potential correlation between the various sphenoid sinus pneumatization types and the sella turcica.

## MATERIALS AND METHODS

The retrospective study with the number (HRÜ/2022/18/31) was approved by the Harran University Clinical Research Ethics Committee. Prior to the study, a power analysis was conducted referring to the study by *Bilgir & Bayrakdar (2021)* to ensure that the obtained data could be appropriately used and evaluated. The Raosoft sample size calculator was used to calculate the required sample size. Considering that 1,000 full-head CTs were requested in the Department of Dentomaxillofacial Radiology over a 2-year period, it was determined that 400 images were needed to be examined to achieve a margin of error of 5% and a confidence interval of 99%. All images were taken according to a standard scanning protocol, using a multi-slice scanner (Revolution; GE Healthcare) with 120 KV, 110 mAs, 25 cm FOV and 0.6-mm slice thickness.

For this study, archival records of all head CT images taken from patients who presented to our hospital with different complaints were retrospectively reviewed, and only the images of individuals who met the screening criteria were included. Any images with artifacts in the study area were excluded. In total, 420 CT images were analyzed *via* multiplanar projections, and they were categorized by age and gender. Radiological scans and examinations were carried out by M.E.D, who has 5 years of experience.

To ensure the reliability of the results, a pre-study assessment was conducted to evaluate intra-observer agreement. The observer evaluated a minimum of 80 CT images, which were repeated after a 2-weeks interval to check for consistency. The classification of sella pneumatization types was based on the conchal, presellar, incomplete sellar, and complete sellar categories. The classification system proposed by *Hammer & Radberg (1961)* was employed to evaluate sphenoid sinus pneumatization, with the sellar type further subdivided into complete and incomplete categories. If the pneumatization extended to the clivus, the incomplete type was included in the complete type, as depicted in Figs. 1–3. The morphology of the sella was assessed using the Axelsson classification (*Axelsson, 2004*).

### Statistical analysis

The analysis of the data gathered in this study was subjected to statistical scrutiny through the use of IBM SPSS 25.0 software package (Armonk, NY, USA). The obtained results were conveyed in terms of the number of units (N), percentage (%), and mean ± standard deviation. To evaluate the normality of data dispersion, the Kolmogorov-Smirnov test was implemented, while the relationship between categorical factors was explored using the Pearson chi-square test. Furthermore, to ensure the consistency of the results, the kappa statistic was employed to evaluate intra-observer agreement.

## RESULTS

In this study, a total of 420 CT images were evaluated for intra-observer agreement, resulting in an excellent agreement score of 0.96. The mean age of the patients included in the study was 43.87 ± 17.58 years, with 174 females (mean age 42.67 years) and 246 males (mean age 44.72 years). While the entire study consisted of subjects with minimum age of 8 and maximum of 91, subjects with minimum age of 8 and maximum of 91 in males and

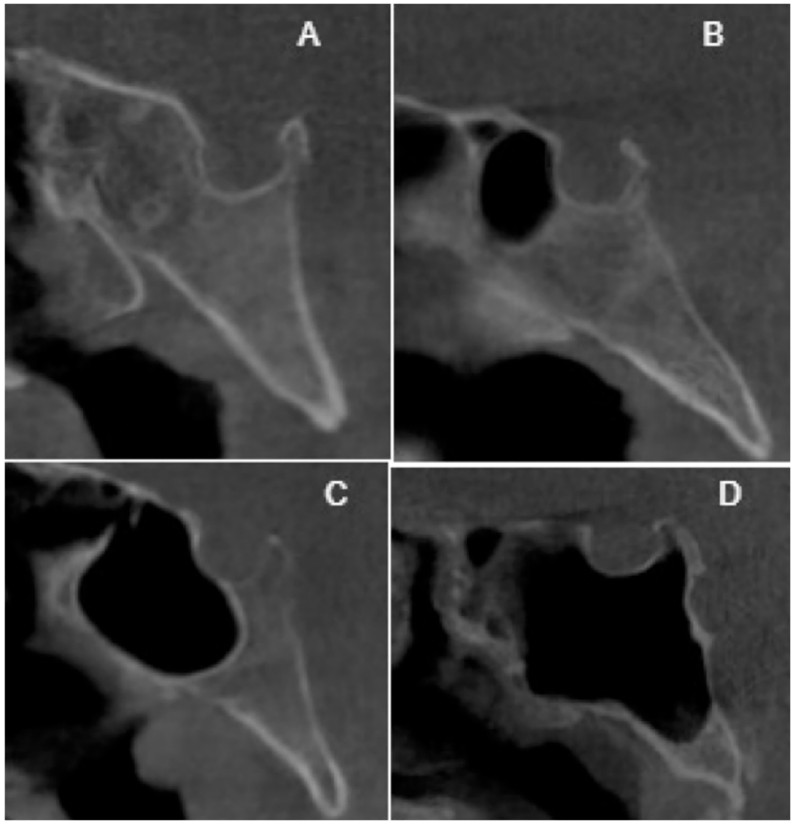

**Figure 1 Cropped CT images in sagittal section conchal pneumatization (A), presellar pneumatization (B), incomplete sellar pneumatization (C), complete sellar pneumatization (D).**

minimum of 12 and maximum of 81 in females were included. The distribution of sphenoid sinus pneumatization by gender is given in Table 1. A statistically significant difference was found between sphenoid sinus pneumatization types according to age groups ($p < 0.05$). Pneumatization by age groups is shown in Table 2. The most commonly observed morphological type of sella turcica was irregularity in the posterior part of the dorsum sella, accounting for 51.2% of cases. Other observed types included normal sella turcica in 41.9% of cases, oblique anterior wall in 2.9%, dorsum sella pyramidal shape in 2.4%, and double contour of floor in 1.7%.

When examining the relationship between sphenoid sinus pneumatization types and sella types, it was found that complete sellar pneumatization was predominantly associated with the irregular sella type, with a prevalence of 26.7%. The prevalence of complete sellar with a normal sella morphology was 19.0%, while complete sellar was most frequently followed by irregularity in the posterior part of the dorsum sella at a rate of 19.0%. Presellar pneumatization was predominantly detected with a normal sella morphology at a rate of 8.8%, while conchal pneumatization was mostly observed in conjunction with a normal sella morphology at a rate of 1.9%. These findings are summarized in Table 3.

A statistically significant relationship was found between sphenoid sinus pneumatization and the morphological types of sella ($p < 0.05$).

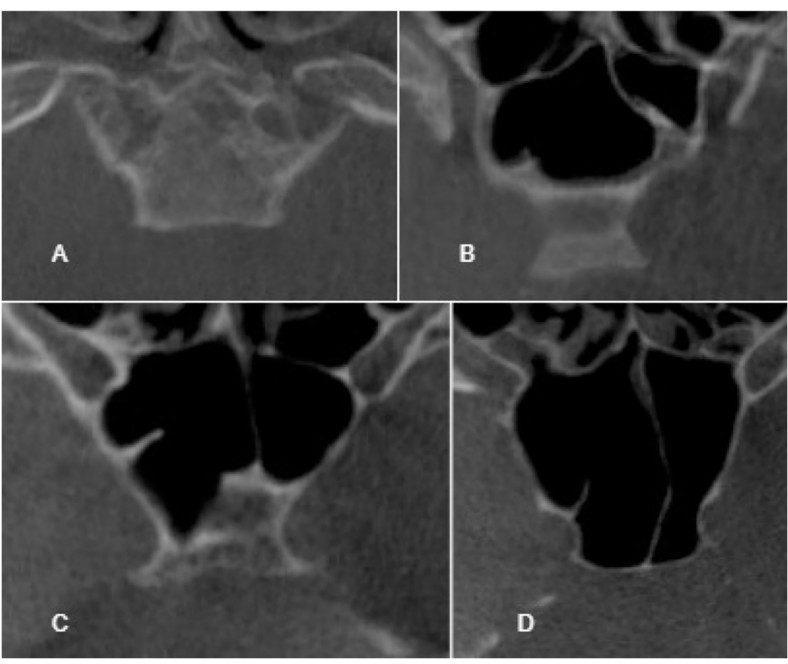

**Figure 2 Cropped CT images in axial section conchal pneumatization (A), presellar pneumatization (B), incomplete sellar pneumatization (C), complete sellar pneumatization (D).**

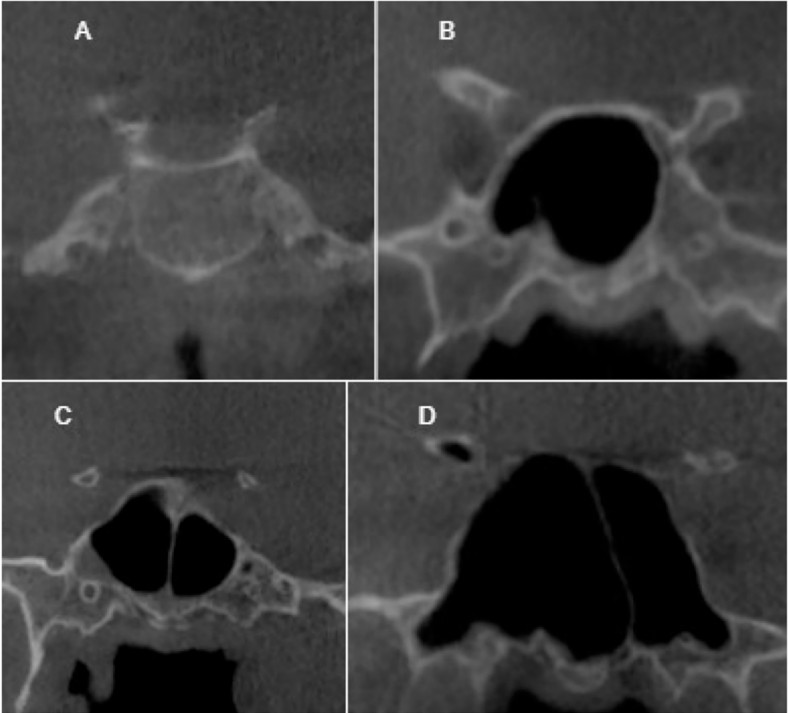

**Figure 3 Cropped CT images in coronal section conchal pneumatization (A), presellar pneumatization (B), incomplete sellar pneumatization (C), complete sellar pneumatization (D).**

**Table 1 Sphenoid sinus pneumatization according to gender.**

| Gender | Sphenoid sinus pneumatization | | | | | |
|---|---|---|---|---|---|---|
| | Conchal N (%) | Presellar N (%) | Incomplete sellar N (%) | Complete sellar N (%) | Total N (%) | p value |
| Male | 6 (1.5) | 30 (7.1) | 82 (19.5) | 128 (30.5) | 246 (58.6) | 0.063 |
| Female | 4 (1.0) | 38 (9.0) | 61 (14.5) | 71 (16.9) | 174 (41.4) | |
| Total | 10 (2.5) | 68 (16.1) | 143 (34) | 199 (47.4) | 420 (100) | |

**Table 2 Sphenoid sinus pneumatization by age groups.**

| Age groups | Conchal N (%) | Presellar N (%) | Incomplete sellar N (%) | Complete sellar N (%) | Total N (%) | p value |
|---|---|---|---|---|---|---|
| 8–20 | 5 (50) | 17 (25) | 21 (14.7) | 18 (9) | 61 (14.5) | 0.008* |
| 21–33 | 0 (0) | 9 (13.2) | 19 (13.3) | 38 (19.1) | 66 (15.7) | |
| 34–46 | 0 (0) | 17 (25) | 28 (19.6) | 42 (21.1) | 87 (20.7) | |
| 47–59 | 5 (50) | 11 (16.2) | 45 (31.5) | 64 (32.2) | 125 (29.8) | |
| 60 years and older | 0 (0) | 14 (20.6) | 30 (21) | 37 (18.6) | 81 (19.3) | |

**Note:**
* $p < 0.05$ exhibits a significant difference.

**Table 3 Distribution of sphenoid sinus pneumatization according to morphological types of Sella.**

| Sphenoid sinus pneumatization | Sella morphological types | | | | | |
|---|---|---|---|---|---|---|
| | Irregularity in the posterior part of the dorsum sella N (%) | Normal sella turcica N (%) | Oblique anterior wall N (%) | Dorsum sella pyramidal shape N (%) | Double contour of floor N (%) | p value |
| Conchal | 1 (0.2) | 8 (1.9) | 0 (0.0) | 1 (0.2) | 0 (0.0) | 0.002* |
| Presellar | 22 (5.2) | 37 (8.8) | 3 (0.7) | 3 (0.7) | 3 (0.7) | |
| Incomplete sellar | 80 (19) | 51 (12.1) | 4 (1.0) | 5 (1.2) | 3 (0.7) | |
| Complete sellar | 112 (26.7) | 80 (19) | 5 (1.2) | 1 (0.2) | 1 (0.2) | |

**Note:**
* $p < 0.05$.

# DISCUSSION

The sphenoid sinus is an intriguing hollow region in the middle of the skull that is posterior to the nasal cavity and filled with air. After birth, it develops and is bilaterally symmetric. The sphenoid sinus is often not pneumatized at the moment of birth. The process of pneumatization, however, begins after the age of 4 years and is completed between the ages of 6 and 12. The surrounding bone grows and changes during the pneumatization process, which results in the formation of a network of air cells inside the sinus. Sphenoid sinus pneumatization is a difficult process that differs greatly from person to person. Age, gender, ethnicity, and environmental factors are just a few of the variables that might affect this process. Therefore, research on sphenoid sinus pneumatization is necessary to comprehend the variety of this process and how it affects people's health. Previous studies have shown that the sphenoid sinus is fully pneumatized in most individuals by the age of 8 (*Terra et al., 2006*; *Cho et al., 2010*; *Wiebracht & Zimmer, 2014*;

*Štoković et al., 2016*). To ensure that the cases selected for this study had completed sphenoid sinus pneumatization, a minimum age of 8 years was chosen. This decision was made because by this age, the sphenoid sinus has typically undergone complete pneumatization, and the air cells have fully developed. This age criterion ensures that the results of the study are accurate and that the sample used for the analysis is appropriate.

Historically, intrasellar lesions situated within the sella turcica have been treated surgically. Among the various surgical approaches, transsphenoidal surgery has emerged as a widely preferred option due to its comparatively lower rates of disease recurrence and mortality when compared to transcranial access. One of the most significant advantages of transsphenoidal surgery is that it offers access to other regions of the skull base through the sphenoid sinus pneumatization, making it easier for the surgeon to reach the lesion (*Wang et al., 2010*). However, it is critical to consider the direction and level of sphenoid sinus pneumatization when deciding on a surgical approach to avoid any iatrogenic harm and to gain a comprehensive understanding of the pathogenesis of the processes occurring within the sinus cavity (*Hiremath et al., 2018*). Hence, it is necessary to have a detailed comprehension of the anatomical variations in the sphenoid sinus.

Several classifications of sphenoid sinus pneumatization have been proposed, including conchal, presellar, and sellar types. The sellar type is particularly noteworthy as it plays a crucial role in determining the transsphenoidal surgical approach. The sellar type can be further divided into incomplete and complete sellar patterns based on the classification system proposed by *Cho et al. (2010)*. This classification helps to determine the degree of sphenoid sinus pneumatization, which is vital in determining the surgical approach for intrasellar lesions. Thus, it is vital for surgeons to have a thorough understanding of the anatomical variations in the sphenoid sinus and its classification to choose the most suitable surgical approach and prevent complications during the treatment of intrasellar lesions. The classification of sphenoid sinus pneumatization is of particular significance in deciding the degree of sellar pneumatization and selecting the appropriate surgical approach (*Battal et al., 2014*; *Gibelli et al., 2017*). We followed the approach of *Cho et al. (2010)* and classified the sellar type into incomplete and complete sellar patterns.

The correlation between the morphology of the sella and sphenoid sinus pneumatization remains unexplored in the scientific literature. While there have been a few studies that have investigated the pneumatization of the sphenoid sinus, these studies have reported inconsistent prevalence rates for the different types of pneumatization. This lack of clarity indicates the need for further research in this area to gain a better understanding of the relationship between the two structures (*Dafalla et al., 2017*; *Degaga et al., 2020*). Such information can be critical in improving the diagnosis and treatment of conditions involving the sphenoid sinus and sella, and ultimately enhance the clinical outcomes of patients (*Yèkpè et al., 2018*). Table 4 summarizes recent studies conducted in different populations.

*Wang et al. (2010)* examined 100 sphenoid sinuses with CT and found presellar and sellar type pneumatization in 2% and 98% of cases, respectively, with no conchal type observed. *Tomovic et al. (2013)* found conchal, presellar, and sellar types in 1.8%, 7.3%, and 90.9% of cases, respectively, while *Lu et al. (2011)* observed the conchal, presellar, and

**Table 4 Comparison of prevalence of sphenoid sinus pneumatization types in different populations.**

| Authors | Population | Imaging (N) | Type of sphenoid sinus pneumatization % | | | |
|---|---|---|---|---|---|---|
| | | | Conchal | Presellar | Sellar | |
| | | | | | Incomplete sellar | Complete sellar |
| *Lakshman, Viveka & Thondupadath Assanar (2022)* | India | CT 52 | 0 | 11.5 | 88.5 | |
| *Bala & Shahdad (2019)* | India | CT 200 | 1 | 9 | 90 | |
| *Hiremath et al. (2018)* | India | CT 500 | 0 | 1.2 | 22.2 | 76.6 |
| *Abdalla (2020)* | Iraq | CT 250 | 0 | 11.2 | 88.8 | |
| *Degaga et al. (2020)* | Ethiopia | CT 200 | 2 | 25.5 | 72.5 | |
| *Ominde, Ikubor & Igbigbi (2021)* | Nigeria | CT 336 | 8.3 | 19.3 | 72.4 | |
| *Dafalla et al. (2017)* | Sudan | CT 506 | 2 | 21 | 76.9 | |
| *Yèkpè et al. (2018)* | Benin | CT 225 | 0.4 | 24.9 | 74.7 | |
| *Bilgir & Bayrakdar (2021)* | Turkey | CBCT 128 | 2.3 | 3.9 | 35.9 | – |
| *Ozenen Keskek & Aytugar (2021)* | Turkey | CBCT 804 | 1 | 5.8 | 26.4 | 66.8 |
| Our study | Turkey | CT 420 | 2.4 | 16.2 | 34 | 47.4 |

sellar types in 6%, 28.5%, and 65.5% of 200 cases, respectively. In another study, *Müderris et al. (2021)* examined the CT images of 113 patients and found conchal, presellar, and sellar types in 1.8%, 7.3%, and 91% (47.7% sellar/43.3% postsellar) of cases, respectively. A study of 59 Korean adults using cadaver heads found the least observed conchal type (1%) and the most common sellar type (90%) (*Cho et al., 2010*).

*Hiremath et al. (2018)* performed 500 CT scans and identified conchal, presellar, incomplete sellar, and complete sellar types in 0%, 1.2%, 22.2%, and 76.6% of cases, respectively. *Bilgir & Bayrakdar (2021)* examined 128 cone-beam computed tomography (CBCT) images and found the conchal, presellar, and sellar types in 3%, 3.9%, and 35.9% of cases, respectively, while in another CT study, the conchal, presellar, and sellar types were identified in 8.3%, 19.3%, and 53.9% of cases (*Ominde, Ikubor & Igbigbi, 2021*). In a study conducted with 804 CBCTs, *Ozenen Keskek & Aytugar (2021)* reported 66.8% complete sellar, 26.4% incomplete sellar, 5.8% presellar, 1% conchal type sphenoid sinus pneumatization. In our study, presellar and complete sellar type pneumatization were found at different rates than this study. These proportional differences may be due to environmental differences and ethnic differences in the regions where the study was conducted, because our study was conducted in the Southeastern Anatolia region and different races coexist in this region.

The results of our study have revealed that the sellar type of sphenoid sinus pneumatization was the most observed, while the conchal type was the least frequent. However, it is crucial to consider the proportional differences among the various types of pneumatization, which may be influenced by factors such as study design, sample sizes, and racial disparities. It is imperative to emphasize the importance of comprehending these variations in sphenoid sinus pneumatization as it can significantly impact surgical planning. Surgeons must consider the different types and degrees of pneumatization to

avoid any iatrogenic harm during surgical procedures. Moreover, a thorough understanding of the pneumatization patterns is essential to comprehend the underlying pathogenesis of the processes occurring within the sinus cavity. This knowledge can lead to better patient care and improved treatment outcomes (*Bala & Shahdad, 2019*; *Abdalla, 2020*).

In the field of anatomical research, computed tomography (CT) imaging is a vital diagnostic technique that offers high-resolution visualisation of the intricate anatomy of the sphenoid sinus and the tissues that surround it. For appropriate surgical planning and clinical care, a thorough understanding of the sphenoid sinus pneumatization pattern is made possible by the use of CT imaging. Therefore, an elementary and important topic of research in the field of anatomy is the investigation of sphenoid sinus pneumatization utilizing CT imaging (*Tomovic et al., 2013*). With the use of CT imaging, scientists can examine the fine features of the sphenoid sinus architecture and how it interacts with other structures. This information is crucial for understanding the pathophysiology of numerous disorders that impact this region, it makes it possible to accurately assess the degree of pneumatization, which is essential for choosing the surgical strategy and avoiding iatrogenic injury. Additionally, the thorough comprehension of the sphenoid sinus pneumatization pattern obtained from CT imaging is a significant advancement in the detection and treatment of a number of conditions, including pituitary tumors, optic nerve compression, and cerebrospinal fluid leaks, greatly advancing clinical practise. Therefore, sphenoid sinus pneumatization CT imaging analysis is a crucial and useful technique in the field of anatomy and clinical practice (*Tahmasbi-Arashlow et al., 2015*; *Lakshman, Viveka & Thondupadath Assanar, 2022*).

The study found that there was a statistically significant difference between the morphology of the sella and the pneumatization of the sphenoid sinus. This implies that there may be an association between these two structures. It is essential to understand the variation in sphenoid sinus pneumatization since it is a complex structure that varies significantly among individuals. Knowledge of this variation is crucial for surgical planning to prevent iatrogenic harm such as injury to the optic nerve or the carotid artery, which can cause severe complications. Understanding the relationship between sella morphology and sphenoid sinus pneumatization can also help in understanding the pathogenesis of processes occurring in the sinus cavity. In our image scan, complete sellar pneumatization can sometimes extend to the posterior of the sella. In the findings of the study, the most irregularity was observed in the posterior of the dorsum sella with complete sellar pneumatization. Thinning can be observed as a result of irregularity in the posterior wall of the dorsum sella. In this case, if complete sellar pneumatization is present, it may cause complications such as perforation and post-perforation during surgery. Future studies should aim to investigate this relationship in more detail, taking into account various factors such as study design, sample size, and demographic characteristics of the study population. Further research can provide insights into the underlying mechanisms that may explain the observed differences and may help develop better surgical strategies for the treatment of sphenoid sinus disorders. Overall, this study highlights the importance of

understanding the complex structures of the sphenoid sinus and the sella and their relationships for effective surgical planning and management of sinus disorders.

## CONCLUSIONS

This study aimed to investigate the prevalence of complete sellar sphenoid sinus pneumatization and irregularity in the posterior part of the dorsum sella in patients undergoing transsphenoidal surgical procedures. As a result, with complete sellar pneumatization, the most irregularity was observed in the posterior wall of the dorsum sella. In complete sellar pneumatization, when irregularity is also observed in the posterior wall of the dorsum sella, it can sometimes lead to perforations if not taken care of during surgery due to thinning of the posterior wall of the dorsum sella. Our results indicate that these CT findings are highly prevalent, albeit with percentage differences compared to previous studies. However, it should be noted that our study was limited to the analysis of CT images from a single center, and thus generalization of our findings should be done with caution. To address this limitation, larger multicenter studies are warranted to corroborate our results and provide more robust evidence regarding the prevalence and characteristics of these CT findings.

## ACKNOWLEDGEMENTS

We would like to thank Prof. Dr. Izzet Yavuz for taking a role as a guide for us in our research.

### Funding

The authors received no funding for this work.

### Competing Interests

Ajinkya M. Pawar is an Academic Editor for PeerJ.

### Author Contributions

- Mehmet Emin Dogan conceived and designed the experiments, performed the experiments, prepared figures and/or tables, and approved the final draft.
- Sedef Kotanlı conceived and designed the experiments, performed the experiments, prepared figures and/or tables, and approved the final draft.
- Yasemin Yavuz conceived and designed the experiments, performed the experiments, prepared figures and/or tables, and approved the final draft.
- Dian Agustin Wahjuningrum analyzed the data, authored or reviewed drafts of the article, and approved the final draft.
- Ajinkya M. Pawar analyzed the data, authored or reviewed drafts of the article, and approved the final draft.
## Ethics

The following information was supplied relating to ethical approvals (*i.e.*, approving body and any reference numbers):

The retrospective study was approved by the Harran University Clinical Research Ethics Committee (HRÜ/2022/18/31).

## Data Availability

The raw measurements are available in the Supplemental File.

## Supplemental Information

Supplemental information for this article can be found online at http://dx.doi.org/10.7717/peerj.16623#supplemental-information.

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
