# Peer review of "Computed tomography-based assessment of sphenoid sinus and sella turcica pneumatization analysis: a retrospective study"

_PeerJ, doi:10.7717/peerj.16623_

## Round 0.1 · original submission · Major Revisions

The reviewers have highlighted certain issues to be addressed in the manuscript. Both reviewers have insisted on the clinical implications of pneumatization.

Reviewer 1 ·

Basic reporting

Article: Computed tomography-based assessment of sphenoid sinus and sella turcica pneumatization analysis
This report evaluates the relationship between sphenoid sinus pneumatization types and sella turcica using computed tomography. Sphenoid sinus pneumatization and its types have been studied in previous studies. In this study, the sphenoid sinus was evaluated in terms of sella turcica morphology. The English of the manuscript is understandable and fluent.

Experimental design

No comment

Validity of the findings

1. The correlation between sphenoid pneumatization types and sella turcica morphology was examined, but its clinical importance was not mentioned enough in the discussion. What exactly is the clinical significance of the morphological types of sella turcica? And what do they mean for EETA?

2. Knowing the locations of the anatomical structures around the sphenoid sinus is important for surgical intervention, but what is the importance of the correlation of sphenoid sinus pneumatization and sella turcica morphology? Could you give some more information?

Additional comments

3. In the results section, the mean ages of the patients are given, but the minimum and maximum ages are not written. In the discussion part, it was stated that a minimum age of 8 years was selected. In the results section, the minimum and maximum age values should be specified as both genders and total.

Reviewer 2 ·

Basic reporting

Clear and professional English is used throughout.

Adequate references are provided but similar studies are already published worldwide in different ethnic and nationalities populations. I am not aware of any study published in the Turkish population. Please provide references, if any.

Please provide CT images in coronal and axial planes along with animations / graphical representations for a better understanding of the topic.

Experimental design

Please add tables showing the type of pneumatization in relation to age and gender.

Validity of the findings

A conclusion should be rewritten in the context of the result of your study and the implication on surgical techniques to avoid complications.

Additional comments

The Ethics committee letter should be translated into English for our understanding.

---

## Round 0.2 · Minor Revisions

Dear author,

The manuscript still has scope to improve in the methodology section. The reviewer has raised pertinent queries about sample size calculation using power analysis and other radiographic details.

Reviewer 1 ·

Basic reporting

1-Although the study was carried out in Turkey, studies with higher sample numbers than those in the table are not shown in Table 4. It would be better if you add the studies in Turkey.

Experimental design

Material Method section:
1- Power analysis was carried out according to the results of which study?
2- Who made the radiographic examinations?
3- What are the FOV measurements of full-head CT?

Validity of the findings

No comment

Additional comments

First of all, thanks for the fixes.
It is seen that only the first author has expertise in dentomaxillofacial radiology. Contributions by other authors to this work should be noted separately.

---

## Round 0.3 · Minor Revisions

The reviewer has pointed out to work on the literature review part by considering to add Turkish articles which would greatly improve the discussion part of this manuscript.

Reviewer 1 ·

Basic reporting

I encourage you to revise your literature review. Because, contrary to what you claim, you do not have the largest sample size among studies like your study. This is not necessary since you are already doing a power analysis. However, I think it will strengthen your discussion writing. I recommend that you reconsider the studies involving the sphenoid sinus conducted in Turkey using CBCT in the Turkish literature (TR indexed journals). In the study conducted in Turkey, which is included in Table 4, sellar type is not divided into two, thus preventing you from making comparisons.

Experimental design

Thanks for the corrections.

Validity of the findings

No comment

Additional comments

Thanks for the corrections.

---

## Round 0.4 · accepted · Accept

Dear author,

Please check for the accuracy of the reference information added during the last revision in the bibliography and in the text.

Reviewer 1 ·

Basic reporting

No comment

Experimental design

No comment

Validity of the findings

No comment

Additional comments

Thanks for the corrections.
The accuracy of the reference information added during the last revision should be checked in the bibliography and in the text.